



# In-Situ Measurements of Cloud Microphysical and Aerosol Properties during the Breakup of Stratocumulus Cloud Layers in Cold Air Outbreaks over the North Atlantic

5  Gary Lloyd[1,2]*, Thomas W. Choularton[2], Keith N. Bower[2], Martin W. Gallagher[2], Jonathan Crosier[1,2], Sebastian O'Shea[1], Steven J. Abel[3], Stuart Fox[3], Richard Cotton[3], Ian A. Boutle[3]

*1. NERC National Centre for Atmospheric Science (NCAS), UK*

*2. Centre for Atmospheric Science, University of Manchester, UK*

10  *3. Met Office, Exeter, UK*

*Correspondence to*: G. Lloyd (gary.lloyd@manchester.ac.uk)



**Abstract.** A key challenge for numerical weather prediction models is representing boundary layer clouds in Cold Air Outbreaks. One important aspect is the evolution of microphysical properties as stratocumulus transitions to open cellular convection. Abel et al. (2017) has for the first time from in-situ field observations shown that the breakup in cold air outbreaks over the eastern Atlantic may be controlled by the development of precipitation in the cloud system while the boundary layer becomes decoupled. This paper describes that case and examines in-situ measurements from 3 more cold air outbreaks. Flights were conducted using the UK FAAM BAe-146 aircraft in the North Atlantic region around the UK making detailed microphysical measurements in the stratiform boundary layer. As the cloudy boundary layer evolves prior to breakup, increasing liquid water paths, drop sizes and the formation of liquid precipitation is observed. Small numbers of ice particles are also observed. Eventually LWPs reduce significantly due to loss of water from the Sc cloud layer. In 3 of the cases, aerosols are removed from the boundary layer across the transition. This process appears to be similar to those observed in warm clouds and pockets of open cells in the subtropics. After breakup, deeper convective clouds form with bases warm enough for secondary ice production, leading to rapid glaciation. It is concluded that the precipitation is strongly associated with the break-up, with both weakening of the capping inversion and boundary layer decoupling also observed.

## 1 Introduction

Cold Air Outbreaks (CAOs) frequently impact the North Atlantic region during winter months. While the synoptic evolution of these events is often well forecast by Numerical Weather Prediction (NWP) models, the extent of resulting stratocumulus clouds (Sc) and the transition from closed to open cellular convective regions is often poorly represented. Some of the reasons for this include the depth of the boundary layer and size of the convective elements in these events are approaching model resolution and the difficulty of parameterizing ice formation processes in these slightly supercooled clouds (cloud bases between ~ -4 and -10 °C ) (Field et al. 2014).

Abel et al. (2017) used in-situ measurements to evaluate cloud resolving simulations of one of the CAOs described in this paper (case 1). Cloud measurements showed a stratiform region composed primarily of liquid drops and low concentrations of ice particles. Down-wind cloud morphology changed rapidly, with shallow cumulus clouds developing, leading to enhanced precipitation rates and secondary ice production. Simulations using the Met Office Unified Model were able to capture some of the cloud properties but unable to reproduce the liquid water path that was observed. Sensitivity tests showed that ice processes were too active in the model, leading to the removal of too much liquid water. It was also found that decoupling of the boundary layer, triggered by the evaporation of precipitation below cloud base, was a key process involved in the transition to open cellular convection. In this paper we extend the in-situ measurements detailed in Abel et al. (2017) to a further 3 cases to examine the similarities between the cloud, aerosol and thermodynamic properties in case 1 and the new cases. The focus in this paper is on the in-situ measurements made during the CAO investigations, and does not extend to model sensitivity tests performed by Abel et al. (2017).



Accurate prediction of CAOs is important as they are common (Kolstad et al. 2009; Fletcher et al. 2016), and bring the potential for severe weather, including heavy rain, snowfall events, damaging winds and lightning. The ability to capture the horizontal extent of low level Sc clouds in these events, together with an accurate representation of the microphysical properties, is also critical to understanding the impact they have on long and short-wave radiation budgets (Curry et al. 1996). While this paper

focuses on synoptic scale weather events, the problem of reproducing mid latitude Sc cloud is an issue in climate modelling, particularly in relation to the Southern Hemisphere (SH) radiation bias, a result of Southern Ocean (SO) clouds not reflecting enough shortwave radiation. Excessive absorbed shortwave radiation biases, particularly over the SO, due to the direct impact of modelled post cold front Sc clouds not containing enough supercooled liquid water, have been described in previous studies (Bodas-Salcedo et al. 2012; Kay et al. 2016)

Fletcher et al. (2016) provided a useful climatology of Marine Cold Air Outbreaks (MCAOs), showing the similarities between these events in both Northern and Southern Hemispheres, which suggests the measurements presented in this paper from CAOs in the NH could be applied to improve our understanding in other regions.

Attempts to model low level boundary layer clouds in Arctic regions frequently find low biases in Liquid Water Content (LWC) values. Klein et al. (2009) compared model simulations of low level single layer clouds with in-situ and remote sensing

measurements from the Mixed-Phase Arctic Cloud Experiment (M-PACE) (Verlinde et al. 2007), finding complex microphysical parameterizations could reproduce observed Liquid Water Paths (LWPs), while simpler schemes led to an underestimation. Morrison and Pinto (2006) found accurate simulation of observed cloud layers was dependent on the microphysics scheme selected. Liu et al. (2011) tested the sensitivity of model simulations to microphysical processes, finding the underestimation of LWCs was most sensitive to the auto-conversion of small ice to snow and the subsequent depletion of

the liquid phase through the Wegener–Bergeron–Findeisen (WBF) process. Switching this process off over the temperature range of the observed cloud layers led to good agreement with in-situ observations. Field et al. (2014) used the Met Office Unified Model to simulate a CAO during January 2010 and the resulting cloud fields over the North Atlantic. The control model under-predicted LWPs and failed to reproduce the stratiform cloud region. Sensitivity tests found improved simulation when shear driven vertical turbulent mixing was allowed to extend to cloud top. Further improvements were found when the

threshold for heterogeneous ice nucleation was changed from -10°C to -18°C and the efficiency of auto conversion of cloud water to precipitation was reduced. . Abel et al. (2017) showed, that even with many of these improvements included in their control simulation, the stratiform LWP was still too low. Field et al. (2017) also showed that 9 leading regional models all struggled to simulate the same Sc cloud layer.

In-situ observations have often focussed on areas of supercooled Sc in Arctic regions (Verlinde et al. 2007; Lance et al. 2011;

Jackson et al. 2012) and report persistent, horizontally extensive regions of cloud consisting of liquid tops that continually precipitate ice. During the Aerosol Cloud-Coupling And Climate Interactions in the Arctic (ACCACIA) campaign, Lloyd et al. (2014) observed similar cloud properties in the Arctic spring, however, in the summer season, the microphysical properties exhibited greater variability, with stable layer clouds sensitive to the development of drizzle and ice formation processes that were observed to reduce LWPs and to break up the cloud.



During the VOCALS campaign (Wood et al. 2011), observations of warm Sc were made over the subtropical southeast Pacific, with a focus on persistent Sc cloud layers consisting of closed cells and Pockets of Open Cells (POCs) (Stevens et al. 2005) that were embedded within overcast and open cellular regions.

Wood et al. (2008) found POCs were most likely to form overnight and at times when aerosol number concentrations were

low, suggesting that drizzle played an important role in the transition between closed and open cell structures. Terai et al (2014) studied POCs in the same region, finding that the ratio of drizzle to cloud water was an order of magnitude larger than in the overcast regions. Wood et al. (2011) also found that aerosol concentrations in the overcast Sc regions were three times higher than in a POC they studied. They concluded that, although they didn't observe the formation of the POC, Sc clouds with high LWPs are inherently unstable due to coalescence scavenging of Cloud Condensation Nuclei (CCN) that exceeds replenishment

rates. Connolly et al. (2013) used Large Eddy Simulation (LES) to examine the impact of gravity waves on the MBL, finding that although the waves do cause additional drizzle formation, the entrainment of warm dry air into the cloud can cause a change from closed cell to a patchy cloud regime. The simulations they performed suggested that CCN scavenging was unimportant in causing cloud breakup. Boers et al. (1996) found evidence that in warm marine Sc over the Southern Ocean, clouds were depleted by the formation of drizzle that scavenged cloud water through collision coalescence, a process that could

be enhanced through depletion of aerosol particles in the boundary layer (Rosenfeld et al. 2008). Others have also highlighted the importance of drizzle formation in model simulations (Boutle and Abel, 2012; Yamaguchi, Feingold and Kazil, 2017) Bretherton and Wyant (1997) found that, although drizzle did play a role in decoupling, a deepening warming process driven by increasing latent heat fluxes was the main driver for decoupling of the mixed layer in their model simulations of subtropical Sc over a warming sea surface.

Overall, research suggests that a combination of physical properties and processes may play a role in the formation, development and eventual break up of Sc cloud layers during CAOs. These include: (i) the dynamical structure of the boundary layer; (ii) surface fluxes of heat and moisture; (iii) Entrainment of dry air at cloud top; (iv) The availability of CCN and Ice Nucleating Particles (INPs); (v) The microphysical structure, which, in part is controlled by (iv); (vi) The development of both liquid and frozen precipitation particles; (vii) the possible influence of gravity waves on (iii) and (vi).

Adequate description of these in models is key to accurately simulating cloud cover during these events and predicting the impact of the severe weather associated with them. A better understanding of the microphysics that sustain these cloud regions and the factors that lead to their breakup will lead to improved numerical weather prediction and subsequently model estimations of reflected shortwave radiation by these boundary layer clouds.

This paper presents in-situ observations during 4 CAOs of clouds deep within areas of Sc, and in the transition regions that are

associated with the eventual evolution to open cellular convection. The main focus is on the microphysics of these cloud layers and the aerosol properties in the boundary layer.

**2.0 Measurements and Data Analysis**



The Facility for Airborne Atmospheric Measurements (FAAM) British Aerospace-146 (BAe-146) aircraft was used to perform a combination of straight and level runs (SLRs) and profiles through cloud layers. Dropsondes were deployed to measure the thermodynamic structure of the atmosphere. In-situ measurements of cloud microphysical and aerosol properties in each case were provided by a suite of instruments. The configuration varied between flights and included a 3 View-Cloud Particle Imager

(3V-CPI, Stratton Park Engineering Company (Spec) Inc. Boulder, USA) consisting of two instruments, the 2-Dimensional Stereoscopic (2D-S) probe, providing 10 µm resolution shadow images of hydrometeors over the range $10 < d_p < 1280$ µm, and a Cloud Particle Imager (CPI) CCD camera that was used to record very high resolution (2.3 µm) particle imagery (cases 3 and 4). A standalone 2D-S (Lawson et al. 2006) was used in cases 1 and 2. Measurements of larger cloud particles were made using a Cloud Imaging Probe-100 (CIP-100 DMT) (Baumgardner et al. 2001), which provided 100 µm resolution

shadow images of cloud hydrometeors over the range $100 < d_p < 6200$ µm.

Total Water Content (TWC) (liquid + vapour) was measured at 64Hz using a Lyman-Alpha absorption hygrometer (Nicholls et al. 1990). A Cloud Droplet Probe (CDP-100 Version-2, Droplet Measurement Technologies (DMT), Boulder, USA) (Lance et al., 2010) was used for measurement of the cloud droplet size distribution over the range $3 < d_p < 50$ µm. Calibration was carried out using glass beads and the data compared to measurements from the Nevzorov Liquid Water Content and Total

Water Content Probe (LWC + IWC) (Korolev et al. 1998), which was baselined following the method in Abel et al. (2014). Good agreement between instruments was found for LWCs measured during cloud profiles. A Microwave Airborne Radiometer Scanning System (MARSS) (McGrath and Hewison, 2001) was also used for column liquid water retrievals. Additional details on the MARSS LWP retrieval can be found in Abel et al. (2017).

Measurements of the aerosol size distribution were made using a Passive Cavity Aerosol Spectrometer Probe (PCASP-100X,

DMT) for particle sizes $0.1 < d_p < 3$ µm (Rosenberg et al. 2012). Core instruments on the aircraft included temperature, measured using Rosemount/Goodrich type 102 temperature sensors and information about aircraft altitude, speed and position was provided by the GPS-aided inertial navigation system.

The shadow imaging probe 2D-S was used for geometric analysis of particle size and shape. From this information discrimination between irregular and spherical particles was possible for hydrometeors > ~ 60 µm using a circularity criterion

(Crosier et al. 2011). The categories generated using information about a particles shape were: Low Irregular (LI, shape factor between 1 and 1.2), indicating liquid droplets; Medium Irregular (MI, shape factor between 1.2 and 1.4), for increasingly irregular particles possibly indicative of ice; High Irregular (HI, shape factor of 1.4 and greater), images that have a high shape factor, indicating ice particles.

Data from these instruments were analysed using the Optical Array Shadow Imaging Software (OASIS), which was developed

by the National Centre for Atmospheric Science (NCAS) and DMT, further description of this can be found in Crosier et al. (2011). The 2D-S and CIP-100 were fitted with Korolev anti-shatter tips (Korolev et al. 2011) to reduce shattering artefacts. Examination of Inter-Arrival Time (IAT) histograms was also used to identify and remove shattered particles. The 2D-S contained within the 3V-CPI instrument used a knife-edge inlet to limit artefacts. Datasets in 1Hz format were used to calculate statistics such as mean liquid water content profiles as a function of altitude, and for percentiles in figures 6, 9 and 12.



**2.1 Microphysical Evolution with Proximity to the Breakup Region**

In cases 1-3 the microphysical properties of the stratiform capped boundary layer were related to the proximity to the transition region to investigate changes in properties of the cloud layers. Forward trajectories using boundary layer winds from the mid-point of each aircraft profile were simulated by the Met Office Unified Model using the same convection permitting model
configuration used in Abel et al. (2017). From this, with the use of MODIS satellite data, the distance each profile was situated from the breakup region was estimated, and some of these trajectories are shown as examples overlaid on satellite imagery in figure 1, with the estimated location of the breakup highlighted by the white boxes. This information was then used to calculate the median and inter-quartile ranges for a range of microphysical properties as a function of the distance to the breakup of the Sc. These are presented in figures 6, 9 and 12 for cases 1, 2 and 3 respectively.
In case 4, profiles through Sc were all made at a similar distance to the breakup as determined by the forward trajectories. For this reason the analysis of changes occurring towards the breakup zone was not carried out for this case.

**3.0 Observations**

Fig. 2 shows ECMWF ERA-5 reanalysis products for surface pressure, 2 metre temperatures, sea surface temperatures and 10 metre wind speeds for four CAOs over the North Atlantic region between 2013 and 2016. Fig. 3 shows flight tracks and
examples of profile locations numbered on MODIS satellite imagery for each case alongside calculated liquid water paths from the CDP and MARSS. The following sections describe the aerosol and microphysical properties of each case.

**3.1 Case Study 1 – 24 November 2013**

This is the case that was analysed by Abel et al. (2017). A blocking anticyclone centred to the west of the UK led to a CAO which was flown in between the North East coast of the UK and the Norwegian coast (ERA5 Fig 2a-d). 10 metre wind speeds
were NNW at about 12 m s$^{-1}$.
 MODIS satellite imagery (fig. 3a), shows Sc cloud developing off the Eastern coast of Greenland, eventually breaking up into open cellular convection towards the Norwegian coastline.  The BAe-146 departed from Prestwick (55.51° N, 4.59° W) before measuring the Sc cloud layers, and eventually open cellular convection. 10 profiles were made in the measurement area, with most of these taking place deep within the Sc layers before a limited number of profiles were made towards the region of open
cellular convection.
In-situ measurements revealed a region of Sc cloud below a temperature inversion between 1750 and 2000m. Dropsonde locations (S1 and S2) are marked on satellite imagery (fig. 3a) and for this case measured dropsonde profiles were in the stratiform region. LWP values measured by the CDP and MARSS (fig. 3a) were generally 200-500 g m$^{-2}$ between 1000 UTC and 1055 UTC (fig. 3a). Profile numbers within the stratiform regions of each case are labelled and coloured red on the LWP
time series in fig. 3, and those determined to be close to the transition are shown in black. In the stratiform region cloud tops were 1750-2000m and cloud bases were variable between ~ 1000 and 1500 m. One profile took place in the Cumulus (Cu)





region where cloud base was about 750 m. The location and temperature of cloud top and cloud bases for each profile are shown on fig. 4a.

Microphysical properties as a function of altitude for cloud profiles are shown in fig. 5. These profiles are examples from the overcast region (P5 and P6) compared with measurements made closer to the breakup region. (P8 and P9). The location of these profiles is also marked on the satellite imagery in fig. 3a.

In the overcast region cloud LWC values measured by the CDP peaked between 0.8 and 1 g m$^{-3}$ (fig. 5a). Cloud profiles closer to the cloud breakup (indicated by visible satellite data) were found to contain lower LWCs (fig. 5c), and eventually the aircraft measured a glaciated convective cell with very little liquid water (fig. 5d). Fig. 3a shows the higher LWPs associated with the stratiform region (P6 and P7 CDP LWP values 499 and 427 g m$^{-2}$ respectively) and the lower values in the transition and open cellular region (P9 and P10 CDP LWP values 108 and 17 g m$^{-2}$ respectively).

Fig. 6 shows microphysical analysis as described in section 2.1. Quoted microphysical properties are median values. Typical droplet number concentrations were around 100 cm$^{-3}$ 220 km away from the breakup deep in the stratiform region. These fell to just 20 cm$^{-3}$ 40 km from the breakup. Cloud droplet sizes were seen to steadily increase from median values of 18 µm 100 km away to 25 µm just 40 km from the breakup.

These concentrations were in broad agreement with boundary layer aerosol measured by the PCASP that were 84 cm$^{-3}$ falling to 27 cm$^{-3}$ ~ 40 km from the breakup (Fig. 6d). Vertical profiles showed these particles were well mixed to cloud base (fig. 5e-h).

Cloud top temperatures were ~ -15 °C (fig. 4a), with little change closer to the breakup region. The cloud base temperature did vary between -10 °C and -4 °C, with the higher temperatures closer to the breakup as shallow Cu was rising into the Sc layer above. Size distributions of cloud droplets were calculated over whole cloud profiles, and a selection from the stratiform region and the transition region are shown in figure 7(a) (P6, green lines and P8, purple lines respectively). The dashed lines represent the very low numbers of highly irregular particles identified by the 2D-S. The size distributions showed a broadening spectrum with increasing proximity to the breakup region. The stratiform region appeared to be dominated by liquid water, with evidence of some ice formation, particularly towards the breakup region. In Profile 8, within the Sc region, there was an increase in concentrations of highly irregular particles and a corresponding decrease in LWP (fig 5c), indicating ice, with peaks around 4 L$^{-1}$ measured by the 2D-S. The ice crystal images from the 2D-S and CIP-100 showed the presence of dendritic ice. (fig. 7a).

The final profile (P9), took place in open cellular convection, just to the SE of the Sc. The aircraft traversed the lower regions of an isolated convective cell. During this profile high concentrations of columnar ice crystals were observed that reached peaks of 120 L$^{-1}$, with a reduction in liquid water as can be seen in fig 5d.

## 3.2 Case Study 2 – 20 November 2013

A CAO affected the North Atlantic region to the North West of the UK (fig. 2e-h), with an area of Sc to the North of the UK that transitioned into open cellular convection to the West of Scotland (fig. 3). A deep area of low pressure produced strong





10 metre winds compared to the other cases in this paper (20 m s$^{-1}$). With the advection of a cold airmass of polar origin significant cloud formed, overlying the warmer sea surface temperatures. Combinations of Straight and Level Runs (SLRs) and saw tooth profiles were performed by the BAe-146 aircraft after departing from Prestwick.

In-situ measurements revealed a region of Sc cloud below the temperature inversion at around 1750 m. Earlier dropsondes
deployed by the aircraft (fig. 4b) had found the inversion to be between about 1500 and 2000m. Dropsonde locations are marked on satellite imagery (fig. 3b). Although both dropsondes were within the stratiform region there is a notable increase in height and weakening of the inversion from the sonde closest to the breakup. LWP values measured by the CDP and MARSS (fig. 3b) were generally 200-600 g m$^{-2}$ in the stratiform region between ~ 1235 UTC and 1255 UTC. Cloud tops were 1750-2000m and cloud bases varied between ~ 1000 and 1500 m. Cloud top temperatures were between -10 °C and -13 °C. Cloud
base temperatures were between -5 and -7 °C. The altitude and temperature of cloud top and cloud bases for each profile is shown on fig. 4b.

Microphysical properties as a function of altitude for selected profiles are shown in fig. 8. These profiles are examples from the overcast region (P5 and P6) compared with measurements made during a profile closer to the breakup region (P12). The location of these profiles is marked on satellite imagery in fig. 3b. In the stratiform region LWC values peaked ~ 0.8 g m$^{-3}$ (fig.
8(a-b)). Closer to the transition, cloud profiles contained similar peak values, but thinner cloud layers. This change is represented in the LWP values of 533 and 393 g m$^{-2}$ in the stratiform region (P5 and 6 respectively marked on fig. 3b) compared with 269 g m$^{-2}$ closer to the transition in P12.

Fig. 9 shows microphysical analysis as described in section 2.1. Quoted microphysical properties are median values. Droplet number concentrations did not show the same trends as in Case 1. There was a general increase in concentrations (~ 60 at 400
km from the breakup to 100 cm$^{-3}$ at 100 km away) and sizes (10 at 400 km to 20 μm at 100km from breakup, respectively) of cloud droplets with increasing proximity to the transition. There was also an increase in aerosol concentrations (~ 30 to 60 cm$^{-3}$ in the boundary layer). The reasons for this are not clear, however, compared to the other 3 cases, wind speed measurements were relatively high, ~ 25 m s$^{-1}$ (compared to 10 m s$^{-1}$ in Cases 1, 3 4) in the boundary layer close to the sea surface. These strong boundary layer winds can be seen in figure 2h. The strength of the winds in this case may have had an impact on
production of sea salt aerosol from the surface.

A selection of PSDs from the stratiform region and the transition region are shown in figure 7b. They showed a small increase in the number of precipitation particles measured by the 2D-S just before the breakup. The number of larger particles, indicating drizzle, increased from 10 to 88 L$^{-1}$ between P6 and P12 respectively, with the number of HI particles, indicating ice crystals, only increasing from 0.1 to 0.2 L$^{-1}$. Imagery showed this to be a mix of supercooled drizzle, graupel type particles and columnar
ice crystals.

**3.3 Case Study 3 – 23 March 2015**

A CAO affected the North Atlantic region east of Greenland (Fig 2i-l) with an area of Sc investigated to the west of Iceland using a combination of SLRs and saw tooth profiles that were performed by the BAe-146 aircraft (Fig. 3c). The stratiform





region associated with this CAO was significantly smaller in area than the other cases presented in this paper, but had similar microphysical properties. Satellite imagery (fig. 3c) showed the Sc region to the west of Iceland eventually breaking up to the SW of the island. Wind speeds at 10 metres were around 10 m s$^{-1}$.

After the aircraft departed from Keflavic (63.99° N, 22.62° W) a high altitude run was carried out with dropsondes (marked
on fig. 3c) used to measure the vertical structure of the atmosphere (fig 4c). These measurements revealed an inversion ~ 1500 m in the stratiform region while a dropsonde in the open cellular region showed that the inversion had been eroded, with associated convection confirmed by satellite imagery.

Cloud top heights in the stratiform region were ~ 1500 m, while 2 profiles closer to the breakup showed increased cloud top height close to 2000 m (fig 4c). Cloud top temperatures were between -10 and -14 °C. Cloud base altitudes were quite closely
grouped around 750 m, but the temperatures varied between -3 and -11 °C. Winds in the boundary layer were North Easterly at ~ 10 m s$^{-1}$.

Initially, as measurements were made with increasing distance south, a change from a shallow layer of Sc consisting of small liquid cloud droplets to a deeper layer containing larger hydrometeors was observed. LWP values measured by the CDP and MARSS (fig. 3c) were generally 200-300 g m$^{-2}$ in the stratiform region ~ 1315 UTC, with values decreasing significantly
closer to the transition from closed to open cell conditions.

Microphysical properties as a function of altitude for selected profiles are shown in fig. 10. These profiles show some examples from the overcast region P2, 4, 5 and 7 compared with measurements made closer to the breakup region (fig. 11) (P9 and P10). In P5 and P7, LWC values measured by the CDP peaked at 0.7 g m$^{-3}$ (fig. 10(c-d)). P9 and 10 close to the transition showed reductions in LWC values to 0.3 g m$^{-3}$ by P10 (fig 11b). LWPs were also seen to decrease to 37 g m$^{-2}$ in P10. P2 and P4 (fig.
3c) are North of where the most extensive cloud layers were observed. In this region the boundary layer appeared to have become decoupled with a double temperature inversion present.

Fig. 12 shows microphysical analysis as described in section 2.1. Quoted microphysical properties are median values. Droplet number concentrations were around 50 cm$^{-3}$ ~ 100 km away from the breakup, but fell significantly to just 15 cm$^{-3}$ about 50 km from the transition. PCASP measurements in the boundary layer were complicated by the presence of precipitation,
however they also show a large reduction from ~ 30 cm$^{-3}$ at 110 km to less than 10 cm$^{-3}$ at around 80 km from the breakup.

A selection of PSDs from the stratiform region and the transition region are shown in figures 7c. Shadow probe and CPI imagery are superimposed on these PSDs as examples of observed particles. The PSDs initially showed a reasonably narrow liquid droplet distribution. As the cloud layer developed this broadened and with increasing proximity to the breakup region (fig 7c), significant increases in precipitation sized particles consisting of drizzle and irregular ice particles were observed.
Despite imagery (superimposed on the size distributions) revealing some irregular particles the concentrations of ice were generally very low, with the number of counts leading to a high degree of uncertainty due to sampling error, therefore no concentrations are stated for this case. During the final profile, drizzle concentrations increased to 50 L$^{-1}$ and images from the CPI and 2D-S revealed spherical liquid drops that were 100-200 µm in diameter.



### 3.3 Case Study 4 – 15 February 2016

A blocking anticyclone ridged across the UK (fig. 2m-p) producing a weak northerly (10 metre winds of 5 m s$^{-1}$) orientated returning polar maritime flow to the north east of the UK, which produced a CAO and a Sc region between the North East coast of the UK and the Norwegian coast. The BAe-146 departed from Cranfield before measuring Sc cloud layers (fig 3d), initially close to the Shetland Isles, but then with increasing distance south along the east coast of England. Profiles through Sc covered a large area from approximately 60 to 61N.

In-situ measurements at the North end of the measurement region (fig. 3d) revealed a region of Sc cloud below a temperature inversion around 1750 m. Earlier dropsondes deployed by the aircraft (marked on fig. 3d) had found the inversion to be between about 1500 and 2000m. The dropsonde profile deeper in the stratiform region had a stronger inversion and lower cloud top height compared to the dropsonde in the transition region. Cloud tops were quite variable in this case and generally between 1500 and 1750 m (fig. 4d). Cloud bases were anywhere from ~1000 to 1500 m. Cloud top temperatures were between -8 °C and -14 °C. Cloud base temperatures were between -6 and -9 °C.

LWP values measured by the CDP and MARSS (fig. 3d) were generally around 200 g m$^{-2}$ in the stratiform region between 1330 UTC and 1445 UTC. Microphysical properties as a function of altitude for selected profiles are shown in fig. 13. These profiles from the overcast region (P3 and P8) compared with measurements made during profiles in an open cellular region identified on satellite imagery (P16 and P18). In the open cellular region a temperature inversion was still present, some profiles exhibited a thin liquid cloud layer below this inversion and others were clear. In the stratiform region, LWC values peaked at ~ 0.8 (fig. 13a-b). During the transition, values were much lower around 0.2 g m$^{-3}$. This change can also be seen in the LWP values (fig 3d) that decrease first to 150 g m$^{-2}$ at the transition and then to around 50 g m$^{-2}$ in the open region. This appeared to be due to the cloud layer either breaking up completely, or the persistence of a thin stratus cloud.

### 4.0 Discussion

ECMWF ERA 5 re-analysis (fig 2) show a range of northerly outbreaks that produces supercooled SCu cloud layers that were investigated over the N. Atlantic during 4 CAOs. These extensive layers of Sc were similar in overall structure, with the cloudy region situated below a temperature inversion around 2 km in a well-mixed boundary layer (fig 14). Visible satellite imagery (fig 1 and 3) shows that all cases eventually transitioned into an open cellular regime, with the Sc cloud breaking up. In-situ measurements by the aircraft were designed to capture the microphysical evolution of the cloudy boundary layer as the Sc transitioned into the open cellular regime.

LWPs in the cloudy region derived in each case through in-situ and remote sensing measurements on the aircraft were generally between 200 and 500 g m$^{-2}$ (fig 3). These values showed a trend to much lower values in the regions of transition (often < 200 g m$^{-2}$).



Case 1 provided observations of decreasing aerosol concentrations in the boundary layer (fig. 6) with proximity to the transition region. This change was independent of the above cloud aerosol, which showed no obvious trend. Formation of drizzle, dendritic and irregular ice particles was observed, with these precipitating from the cloud layer.

It is thought the changes in aerosol below cloud were due to removal of aerosol particles by precipitation scavenging processes.
The removal rate, in the absence of any significant source of replenishment leads to larger cloud droplets, and increased efficiency of the warm rain process, further enhancing precipitation and loss of water from the cloud, which Abel et al. (2017) found to be a key factor in the transition and breakup of the clouds during these events.

Case 2 involved a CAO to the NW of the UK.  This case was in contrast to the other 3 cases, as aerosol properties in the
boundary layer did not show an overall trend of declining number concentrations and actually demonstrated increasing concentrations of aerosol and cloud droplet numbers closer to the breakup (fig. 9). In Case 2 the main difference in the meteorological conditions was the presence of high winds in the boundary layer, in excess of those observed in the other cases. It is proposed that in this situation replenishment of the aerosol population occurred from the sea surface, something that did not happen as much in the other cases. The production of sea salt over the oceans has been found to be strongly dependent on
the wind speed close to the surface (Grythe et al. 2014). This has direct implications for cloud microphysical properties as the droplet size distribution and potentially the availability of IN is modulated

Case 3 provided observations of a CAO off the West coast of Iceland. Sc cloud layers, consisting predominantly of liquid and low concentrations of ice were observed to deepen, with increasing LWPs. This led to the production of larger cloud hydrometeors promoting formation of precipitation and the removal of increasing amounts of water from the cloud. This can
be seen in fig. 12a, where droplet sizes measured by the CDP increase with proximity to the breakup region. The aerosol measurements show that BL concentrations are decreasing with distance south, closer to the breakup, and that cloud droplet sizes are increasing, which would in turn increase the efficiency of the warm rain process.

Case 4 revealed Sc layers with varying amounts of drizzle and irregular ice particles deep within the Sc layer, but much less ice further south and with drizzle almost completely dominating the precipitation. As with Cases 1 and 3, aerosol
concentrations were observed to decrease in the boundary layer with distance travelled south.

In all cases the Sc clouds were situated over a temperature range that was not ideal for secondary ice production to take place through the H-M process. Generally cloud bases, particularly further north, were at the lower end of the H-M temperature zone leading to inefficient production of ice splinters through riming. In some cases, secondary ice production was observed, for example in case 1; instability in the boundary layer allowed cumulus to grow through the H-M temperature zone, penetrating
the Sc cloud layers above. In this case high concentrations of ice were observed, with the rapid depletion of liquid water.

Together with the cloud not being in an ideal environment for H-M secondary ice production, the clouds were also generally warmer than -15 ºC, the temperature at which most dust becomes significant as active INP (Hoose and Mohler, 2012). This led to limited amounts of primary ice nucleation, and when it did occur, the riming particles that were generated descended through a temperature range not conducive to secondary ice production.



Case 1 discussed in this paper describes the CAO investigated by Abel et al. (2017) through both in-situ measurements and modelling sensitivity tests. We have found in a further 3 cases in the N. Atlantic region, that in-situ measurements of their microphysical properties are broadly similar. Predominantly liquid cloud layers contained LWPs of several 100 g m$^{-2}$ before reducing significantly as the cloud evolved into an open cellular regime.

We also confirm the observation of low numbers of ice crystal concentrations, the understanding of which is key to accurately simulating these clouds in NWP models. In 2 of the additional cases, we also observed the removal of aerosol particles from the boundary layer with increasing proximity to the breakup region. In the other new case, this trend was not observed and we can only speculate that higher wind speeds in the boundary layer helped to replenish aerosol from the sea surface.

Abel et al. (2017) provided evidence for boundary layer decoupling in Case 1 by calculating the gradients in ice liquid potential

temperature ($\Delta\theta_{il}$ °C) and total water content ($\Delta q_t$ g kg$^{-1}$) between the surface and top of the boundary layer. They found evidence that total water mixing ratios and liquid potential temperature were both well mixed to the top of the boundary layer. Closer to the transition region increasing values of $\Delta\theta_{il}$ and $\Delta q_t$ were observed indicating drizzle induced decoupling of the boundary layer. Abel et al. (2017) also investigated the potential for cloud top entrainment instability (CTEI) following calculation of parameter $k$ detailed in Lock (2009). Values of $k > 0.2$ are indicative of entrainment at cloud top is conducive to

generating negatively buoyant mixtures that sink, promoting a break up of the cloud. We also calculated this parameter finding the values to be similar to values found by Abel et al. (2017) suggesting CTEI was not a driver for the breakup in these cases. We also carried out the boundary layer decoupling calculations and these have been applied for the cases presented here (fig. 14). We also find evidence for the process of drizzle driven decoupling in some of the other cases. Red profile markers represent gradients in the stratiform region for each case, and black markers in the transition or open cellular regions. Case 2 was quite

different from the other cases in its microphysical evolution and no real relationship is seen. The reasons for this are uncertain, but the strong wind speeds in this case would increase shear induced turbulence, which could keep the boundary layer fairly well mixed, potentially slowing the transition. This well mixed profile would help prevent the decoupling mechanism discussed in case 1 and would also replenish aerosol, reducing the efficiency of any washout-precipitation feedback.

In case 3 there was a lack of suitable measurements in general, but the profile closest to the transition did have increased

gradients. In Case 4 we found strong evidence for this boundary layer decoupling metric being associated with the transition and open cellular regions.

## 4.0 Discussion

In-situ measurements of four CAOs over the North Atlantic were conducted to investigate the microphysical structure of Sc cloud layers and aerosol concentrations in the boundary layer. In all cases, measurements were continued through to the

transition region where cloud layers became disrupted, leading to open cellular convection. The main findings from this study are outlined below.

- Extensive regions of Sc cloud below a temperature inversion were observed in all cases, with these layers eventually breaking up into open cellular convection.



- The further 3 cases studied over the Abel et al. (2017) case are broadly similar in relation to their structure and evolution. This suggests that the results and model improvements detailed in Abel et al. (2017) can be applied more widely to help improve NWP of these types of events.

- The cloud layers in all cases were about 1 km in depth, with cloud top temperatures often warmer than -15°C, with cloud bases colder than or at the colder end of the Hallett-Mossop SIP temperature range (between -3 and -8 °C). Cloud temperatures increased with distance south. This sometimes led to increases in concentrations of columnar ice crystals as often associated with secondary ice production through the H-M process.

- In 3 cases, aerosol properties in the boundary layer showed evidence of a decline in concentration (independent of the above cloud aerosol numbers) with increasing proximity to the transition zone. This has similarities to findings from VOCALS in the eastern southern hemisphere Pacific, where precipitation formation through collision coalescence led to removal of aerosol particles at a rate that exceeded any replenishment. This creates a positive feedback where cloud droplet size increases, together with the efficiency of the warm rain process and eventually the formation of precipitation. This feedback depletes the cloud of water, leading to cloud breakup.

- One case exhibited no apparent trend in aerosol concentrations in the boundary layer. This case was interesting as winds close to the surface were much higher than in other cases. One hypothesis is that replenishment of aerosol from the sea surface due to high wind speeds in the boundary layer offset any losses due to precipitation.

- Precipitation development appeared to be a key process in the evolution of the Sc cloud layers and their eventual breakup. All cases developed significant amounts of drizzle that precipitated from the cloud layers.

- Key to the formation of drizzle precipitation in some cases was the increasing size of liquid droplets measured by the CDP as shown in fig. 6 and fig. 12.

- The ice phase did vary between cases, with generally low concentrations of ice (a few L$^{-1}$) present that appeared to be dependent on cloud top temperature, and whether clouds occupied a significant proportion of the H-M temperature range. Cases 3 and 4 contained relatively little ice, however cases 1 and 2 contained higher concentrations of ice, particularly case 1, where dendritic ice crystals were observed deep within the Sc region, and eventually high concentrations (120 L$^{-1}$) of columnar ice crystals developed as the boundary layer became more unstable and cloud spanned the H-M temperature zone.

- Evidence was found to support the findings of Abel et al. (2017) that small gradients in $\Delta\theta_l$ and $\Delta q_t$ are associated with the stratiform, well mixed boundary layer. Larger values of these parameters were more likely to be found in transition regions, indicating a decoupled boundary layer. It is suggested that this decoupling of the boundary layer close to the break-up zone inhibited the transport of water vapour from the sea surface to the cloud to offset losses

*Acknowledgements.* Airborne data were obtained using the FAAM BAe-146 Atmospheric Research Aircraft, which was operated by Airtask and jointly funded by the UK Natural Environment Research Council (NERC) and the Met Office. We



acknowledge support from NERC under grant NE/I028696/1 as part of the Aerosol-Cloud Coupling and Climate Interactions in the Arctic (ACCACIA) project.

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





**Figure 1:** MODIS visible satellite RGB composite images for each case with flight track of the aircraft overlaid (pink lines). Overpass times are 1200, 1220, 1310 and 1110 UTC respectively. Simulated forward trajectories from the Met Office Unified Model are represented by solid circle symbols, each circle represents a 15 minute time step along the trajectory. White boxes indicate determined cloud breakup location.



5    **Figure 2: ECMWF ERA5 re-analysis data from each case for mean sea level pressure, sea surface temperatures, 2 metre temperatures and 10 m wind speed for case 1 (a-d), case 2 (e-h), case 3 (i-l) and case 4 (m-p).**





**Figure 3: LWP time series values for cases 1-4 (left panels) derived from MARSS (green trace) and CDP (grey markers). Red profile numbers indicate stratiform regions and black markers indicate transitioning or open cellular regions. MODIS RGB composite satellite imagery (right panels) with flight track superimposed (light red trace) and profiles marked (yellow boxes with profile numbers)**





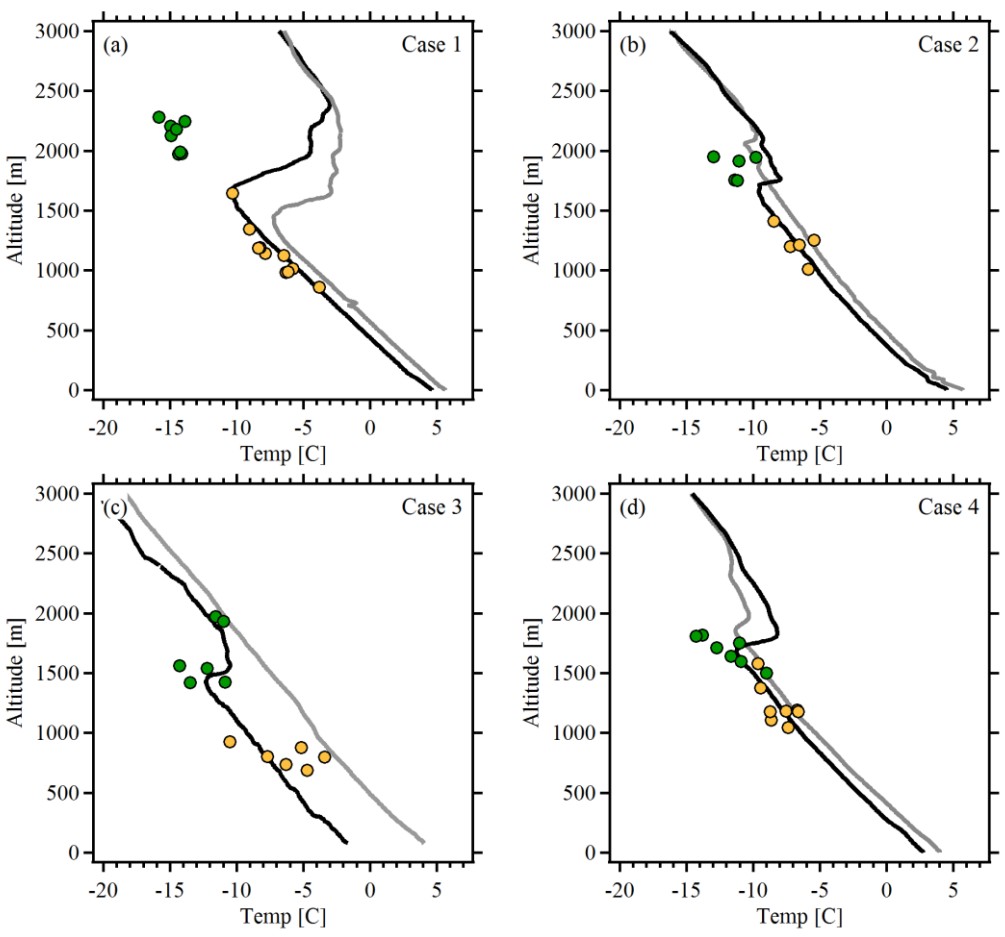

**Figure 4: Cloud top altitude and temperature (green symbols), cloud base altitude and temperature (yellow symbols) and dropsonde data for cases 1-4 showing temperature as a function of altitude. Grey traces represent sonde profiles closer to or in the open cellular regime; black traces represent the stratiform region.**




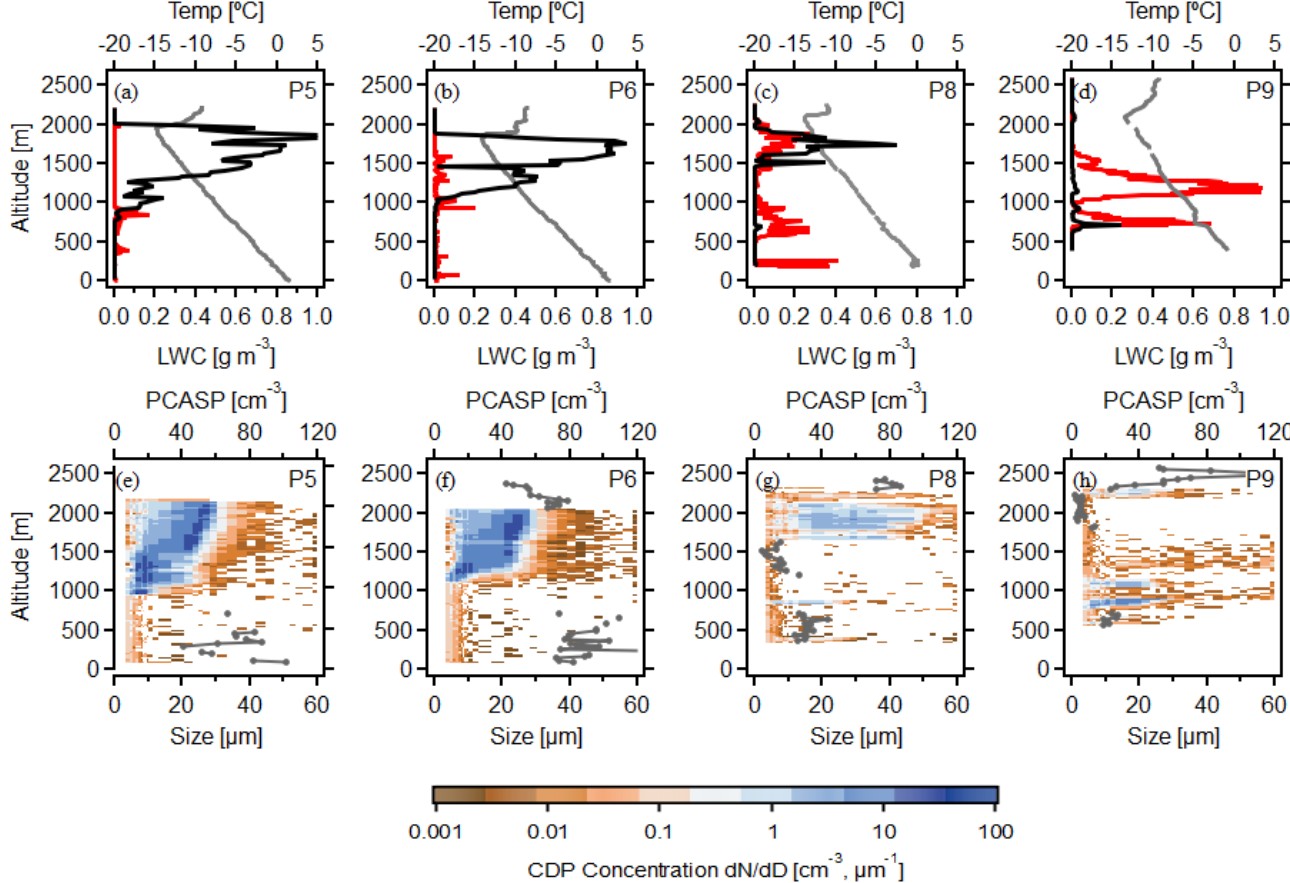

**Figure 5: Measurements from case 1 showing mean LWC as a function of altitude from the CDP (black trace), IWC from the Nevzorov (red trace) and temperature (grey trace) as a function of altitude (a-d). Panels (e-h) show CDP droplet size distributions (colour plot) and mean PCASP concentrations out of cloud as a function of altitude (grey trace.**



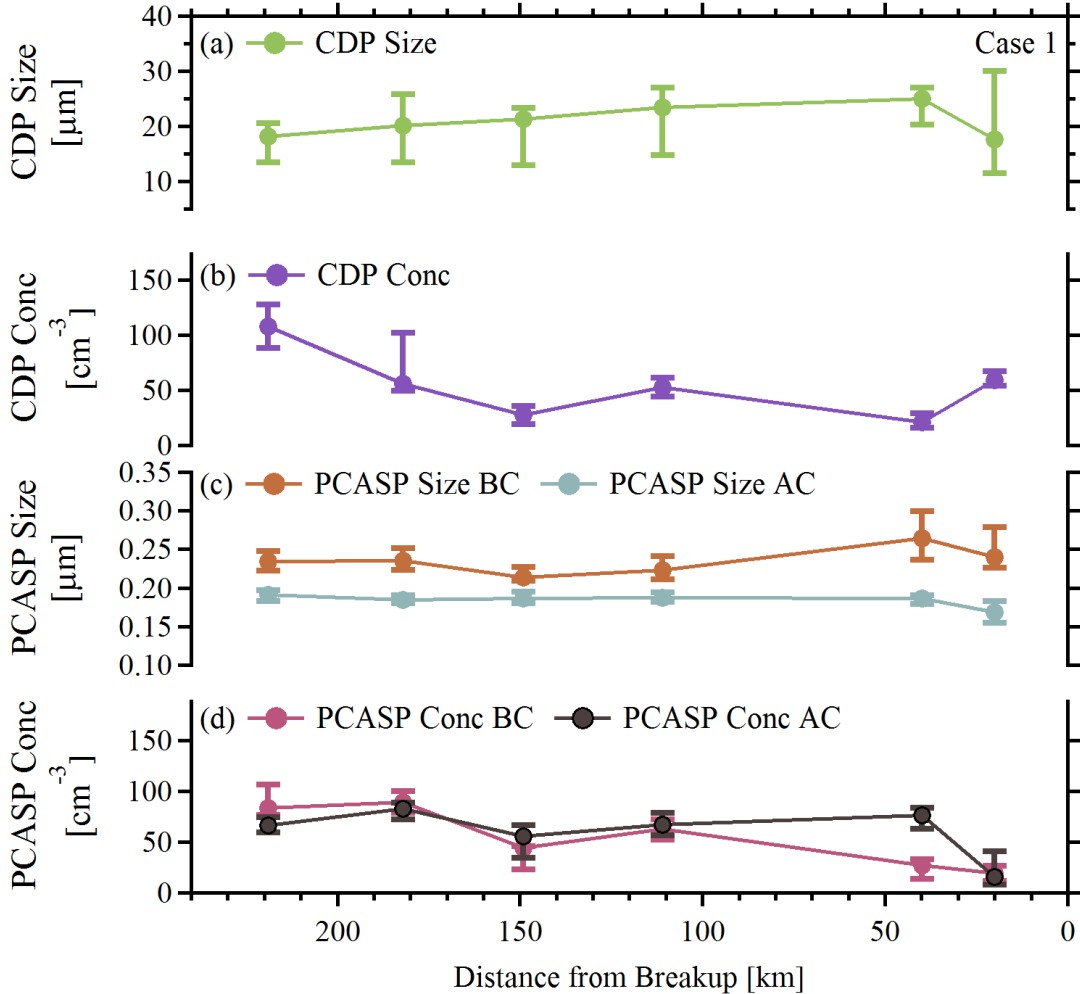

**Figure 6: Median, 25th and 75th percentile values as a function of distance from breakup for (a) CDP size, (b) CDP concentration, (c) PCASP size below cloud (BC) and, above cloud (AC) and (d) PCASP concentration BC and AC.**





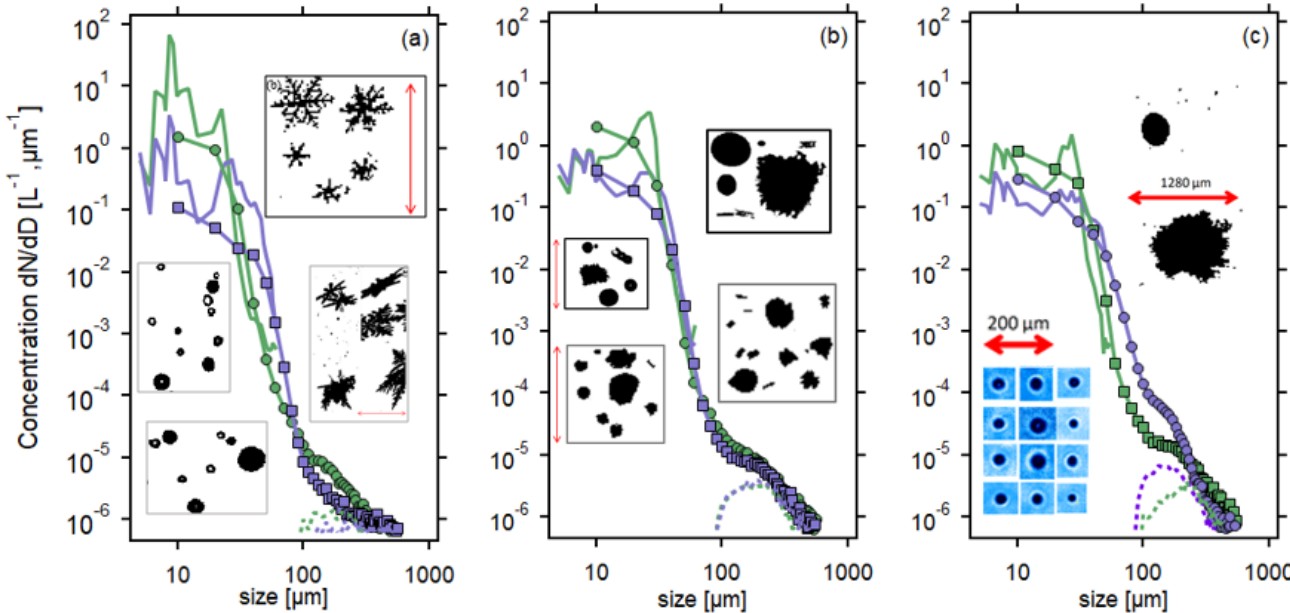

**Figure 7: Size distributions for cases 1-3 comparing the CDP and 2DS size distributions for cloud profiles within the cloudy region (green lines) and closer to the transition (purple lines). Images are examples of ice particles and liquid droplets/drizzle from profiles represented by PSDs.**





**Figure 8: As Fig. 5 but for case 2.**



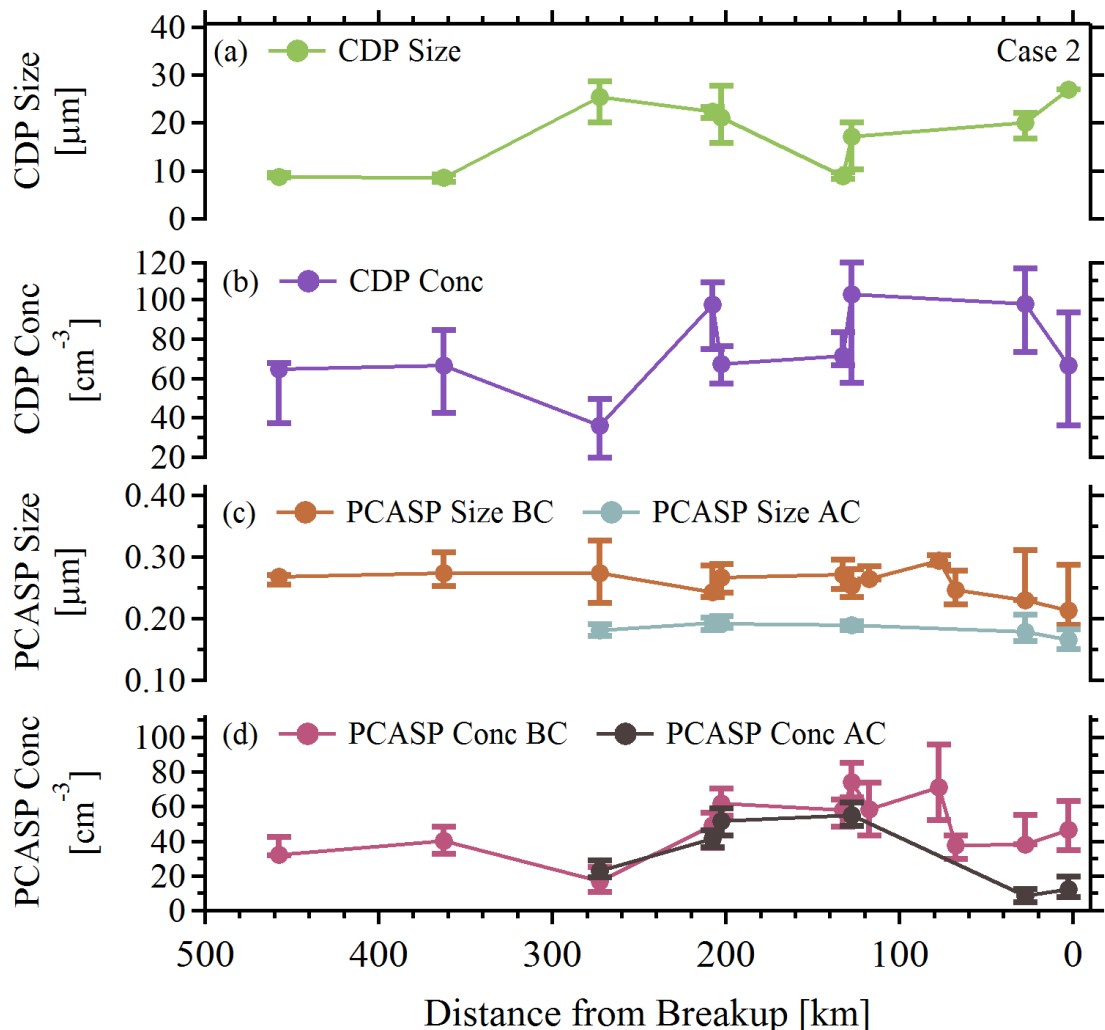

**Figure 9: As fig. 6 but for case 2.**



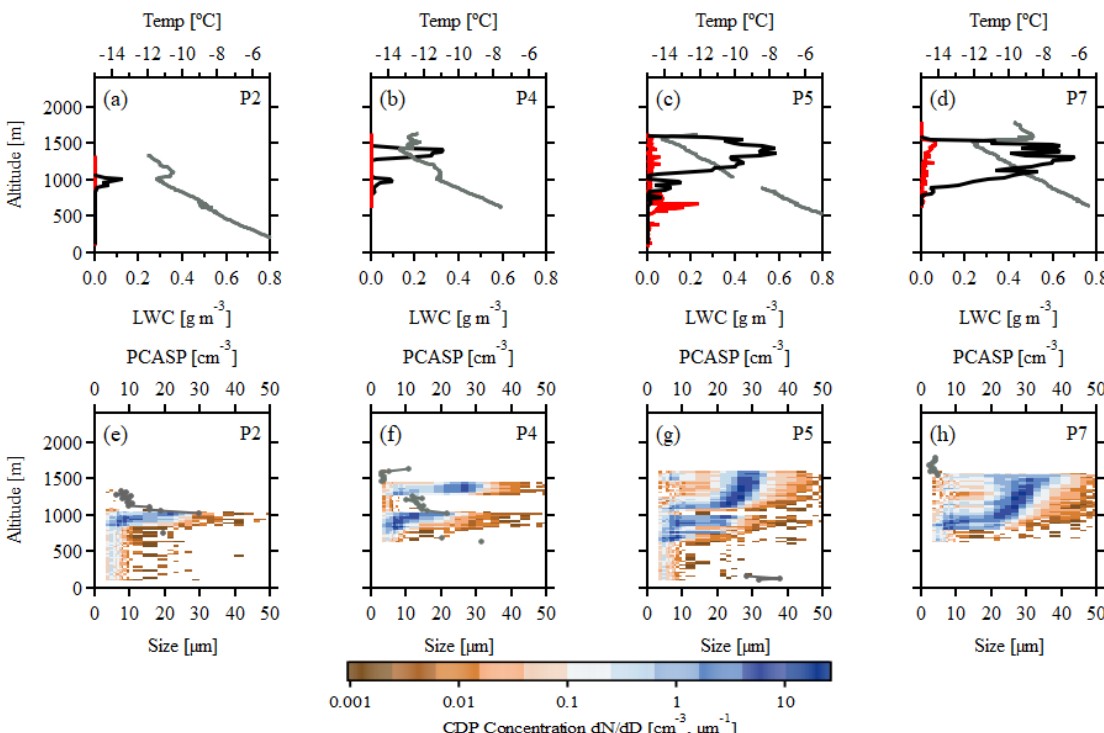

**Figure 10. As fig. 5 but for profiles 2, 4, 5 and 7 for case 3.**



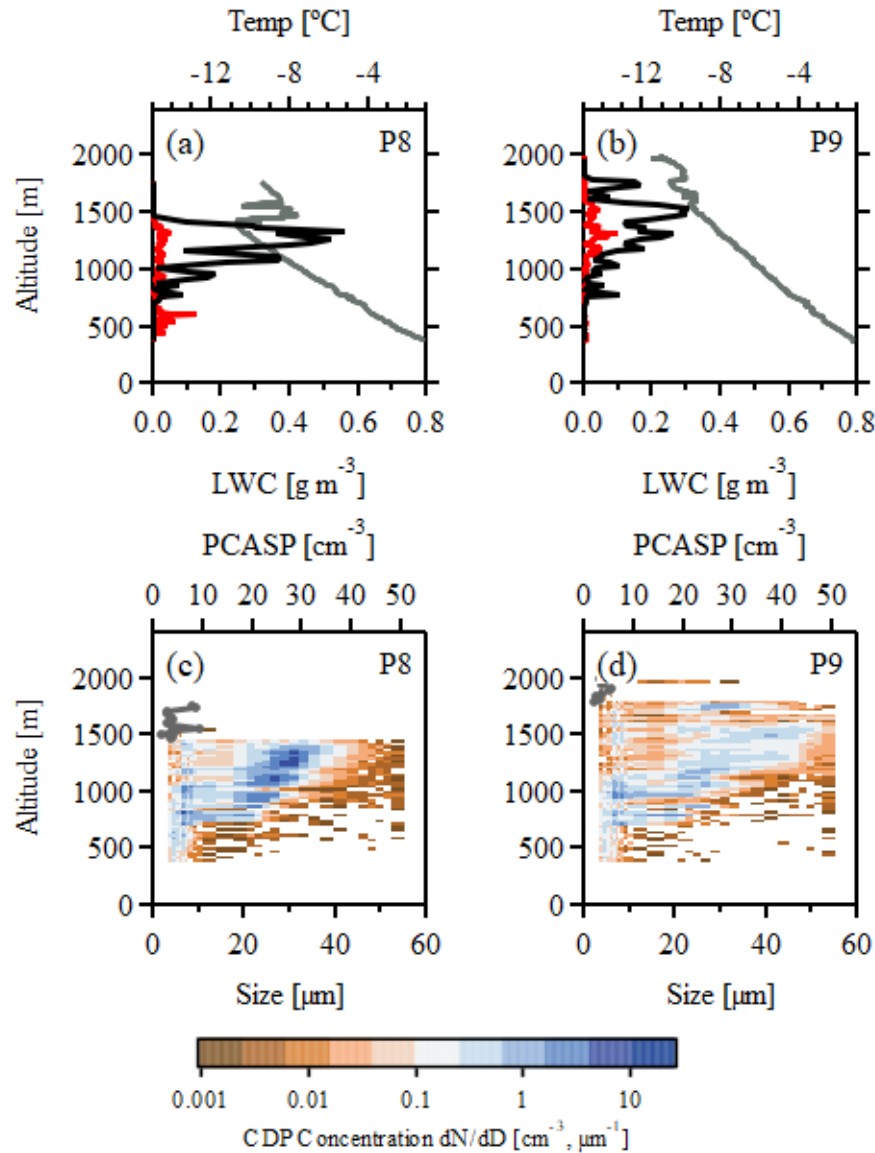

**Figure 11: As Fig. 5 but for profiles 8 and 9 for case 3.**




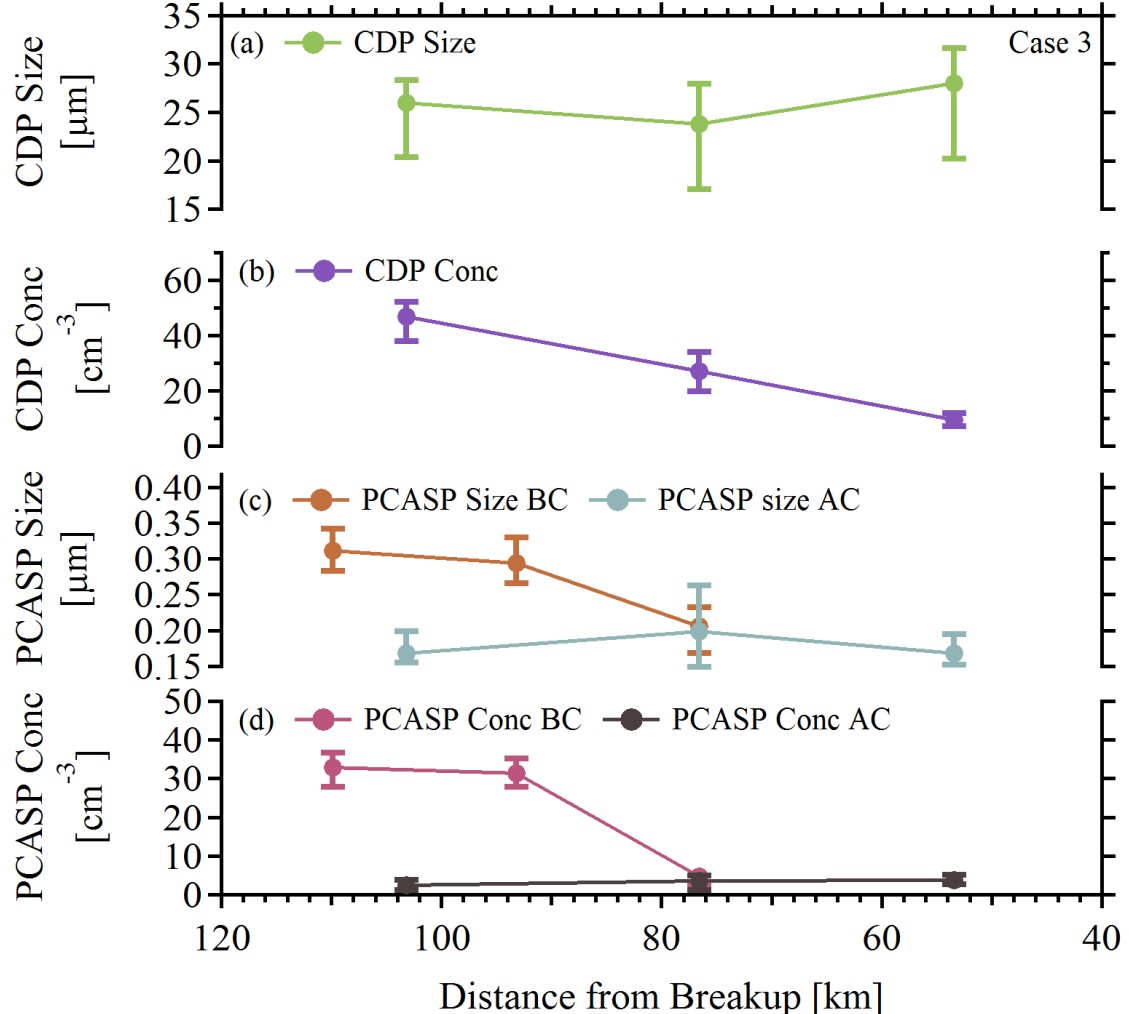

**Figure 12: Median, 25th and 75th percentile values as a function of distance from breakup for CDP size, CDP concentration, PCASP size below cloud, PCASP size above cloud, PCASP concentration below cloud and PCASP concentration above cloud (case 3).**



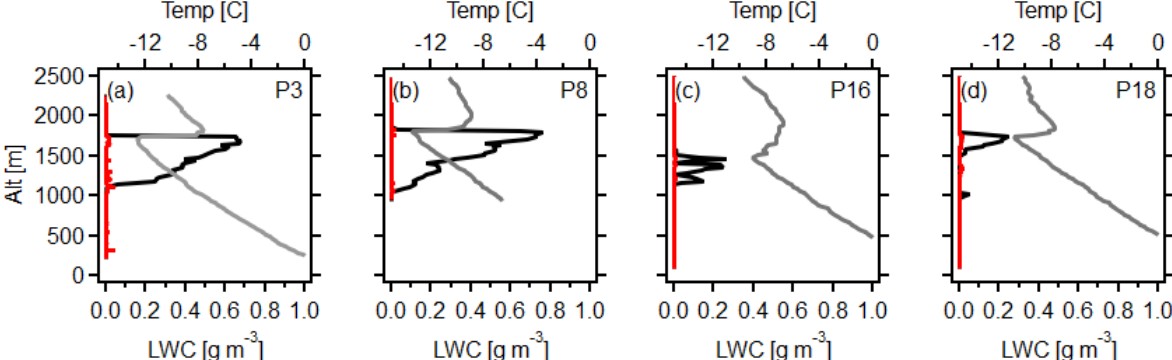

**Figure 13: LWC (black trace) and temperature (grey trace) as a function of altitude for selected profile from case 4.**




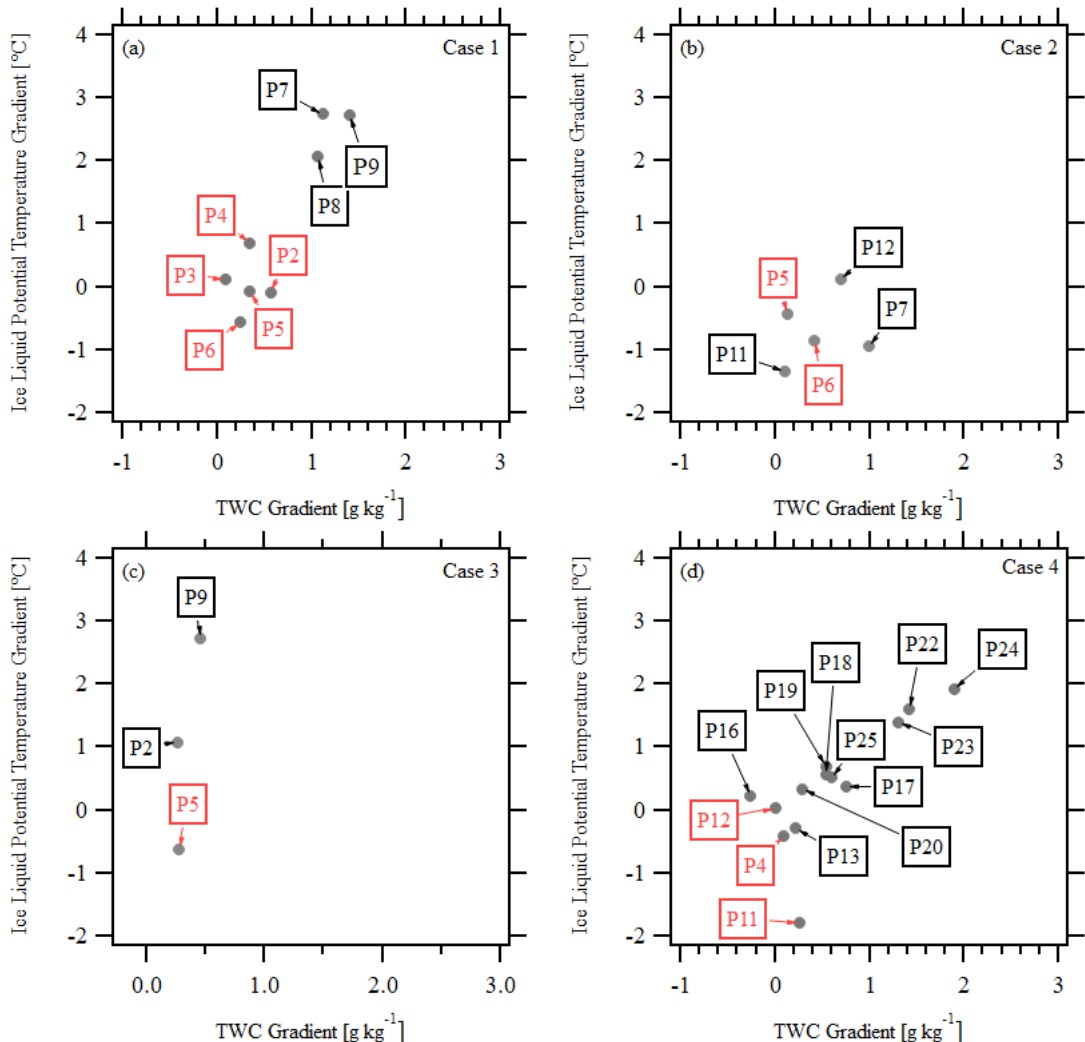

**Figure 14: Gradients in $\Delta\theta_l$ vs $\Delta q_t$ for cases 1-4. Red profile number represent gradients within the stratiform boundary layer and black profile numbers represent gradients in transition or open regions.**