# Peer review of "In-Situ Measurements of Cloud Microphysical and Aerosol Properties during the Breakup of Stratocumulus Cloud Layers in Cold Air Outbreaks over the North Atlantic"

_Atmospheric Chemistry and Physics, 2018_

## Referee Comment (RC1) · Anonymous Referee #1 · 7 Jul 2018

This work examines in situ data from multiple cases of cold air outbreaks over the eastern Atlantic with a focus on their breakup and link with development of precipitation and decoupling of the boundary layer. The authors specifically focus on a case first presented from an earlier study (Abel et al., 2017) and then enhance the work by adding three additional cases. This topic is of importance for the research community interested in boundary layer clouds improving their treatment in numerical weather predictive models. The paper is written well and logically organized. The findings are supported by the analyses presented quite well. Overall, I found this to be an easy

manuscript to read with important findings. I recommend publication and only have minor comments below.

Specific Comments: I believe the authors accidentally labeled two sections as "4.0 Discussion". the second one likely should be "5.0 Conclusions"

Figures: I am unclear about what is meant by "PCASP size" in some of the figures such as Figure 6c and analogous figures for the other cases. How is size calculated? Please provide some discussion.

---

## Referee Comment (RC2) · Anonymous Referee #2 · 28 Jul 2018

Review of "In-situ measurements of cloud microphysical and aerosol properties during the breakup of stratocumulus cloud layers in cold air outbreaks over the North Atlantic" by Lloyd et al.

Recommendation: Major revision

paper presents observations from 4 cold air outbreaks, describing the evolution of the cloudy boundary layer prior to the breakup of the stratocumulus in cold air outbreaks. The authors show that the processes are similar to those observed in warm clouds

and pockets of open cells in the subtropics, and conclude that precipitation is strongly associated with the breakup as is the weakening of the capping inversion and the decoupling of the boundary layer. As the paper presents some unique observations of a phenomenon that has not been well observed, the paper makes a contribution to the literature and should be published. However, there are a few points that the authors should consider in a revised manuscript before the paper is published, particularly as relates to the presentation of the results.

Major Comments

It would have been nice if more attention could have been devoted to a description of what causes some of the differences between the 4 cases that are discussed. For example, was there any variation in the strength of the cold air outbreak between cases and could this be responsible for any of the differences between the cases that were observed. In addition, it would be nice if there was a better way of presenting integrated analysis from all of the different cases rather than just describing the cases 1 by 1 and then having a subsequent discussion. Figure 14 does a good job at this, but it would have been nice to have had more such figures throughout the paper describing other quantities.

Second, I think the nature of the wording of the discussion should be changed. In-situ observations provide measurements of cloud properties, but they do not by themselves measure cloud processes which instead are inferred from the observations. Thus, the statements that the authors make about things being definitively caused by should be replaced by statements like the observations are consistent with . . ..

Third, I think it would be good to provide particle images from more cases than just case 1. This would allow the reader to better assess the statements that are made on the importance of secondary ice crystal production processes throughout the manuscript.

Specific Comments:

Page 2, line 8: Can you replace small numbers with a more specific number?

Page 2, line 22: Can you quantify what you mean by low concentrations?

Page 3, line 26: double period present.

Page 3, lines 29-34: Although this paper focuses on microphysical processes, I think it should be acknowledged that there are many other processes that affect cloud properties. In addition to aerosols, the importance of the lower boundary condition is important as the microphysics will depend on the fetch from the ocean surface.

Page 5, line 4: There could be a problem in comparing measurements from different instrument suites as prior studies have shown that not only the instruments themselves, but also the way the data are processed cold affect the results.

Page 5, line 4: Was there any shattering problem with the 3-V CPI due to the use of the tube? Heymsfield and collaborators have reported that they believe the signal of the 3-V CPI was dominated by shattering on the tube on the 3V CPI. Thus, something should be done to justify the use of the 3V CPI and show whether or not shattering affected the recorded data.

Page 5, line 8: Comparing results from the 2DS on the 3V CPI with a stand-alone 2DS may be problematic because the first may be affected by a shattering problem.

Page 5, line 16: Can you be more quantitative rather than saying good agreement?

Page 5, line 26: Should the thresholds be size dependent, since the number of pixels present also affects some of these shape factors.

Page 5, line 31: Suggested equipped with or outfitted with rather than fitted.

Page 6, line 12: How does the strength of the CAOs for the cases selected vary with the strength of cold air outbreaks that are usually observed in this region?

Page 6, lines 24-25: How were the locations of the profiles selected? This is important

to determine whether there is a bias because of the limited number of profiles that are available.

Page 7, line 3: One of my major comments stated that the comparison of different cases was not as well done in this manuscript as it could have been. Would it be worthwhile to compare the different cases as a function of normalized altitude, because this would allow you to see the vertical profiles from the different cases on a similar scale? That might also help with the organization of the paper as rather than merely documenting the different cases in this section, you could focus your discussion on the different processes and how they vary between the different cases.

Page 7, line 11: Rather than saying typical can you give the mean or median and some indication of the spread.

Page 7, line 22: Can you be more quantitative when you state low numbers?

Page 8, line 1: can you give the index/strength of the CAO for all the different cases, so the reader can see if there is a varying intensity and determine if this might be causing some of the variation between the cases.

Page 9, line 16: How were the "selected" profiles chosen? It would be good to comment on this so that their representativeness is known.

Page 11, line 18: Rather than saying "led to" can you say is "consistent with". See major comment above.

Page 11, line 27: It would have been nice to see more particle images for the different cases rather than just for one case. This would help better illustrate the types of ice crystals present and help determine whether the types of crystals needed for secondary ice crystal production were present.

Page 11, line 32: Do we really know for sure for these cases that the dust are becoming active at -15C? It might have been shown for some past studies but it does not necessarily apply here.

Page 13, lines 6-7: I think with more particle images (rather than for just one case) you could better illustrate what the concentrations of columnar crystals are and better justify this conclusion in the manuscript.

Page 13, lines 14-16: It would be nice if more could be added to the paper to show more integrated analysis of the cases and determine the extent to which varying aerosols, meteorology or strength of cold air outbreaks causes some of the differences between cases.

Page 13, line 18: Can you do a better job of quantifying the concentrations of drizzle in the manuscript if you are making comments about the role of drizzle in the evolution of the system?

Page 19, it would be nice to have date and time labels in the legend or caption.

Page 23: Suggest that you show effective radius and drizzle concentrations in this figure also given some of the conclusions that you are making in the paper.

―――――――――――――――――

---

## Author Comment (AC1) · 26 Sep 2018

Authors Response to Referee Comments on "In-Situ Measurements of Cloud Microphysical and Aerosol properties during the Breakup of Stratocumulus Cloud Layers in Cold Air Outbreaks over the North Atlantic"

We thank the referees for their useful comments on the manuscript. The responses to these points are detailed below.

**Response to Anonymous Referee #1**

Specific Comments

1. The section labelling has been corrected.

2. The PCASP Size is the median particle size that we calculated from the particle size distributions (PSDs) measured by the PCASP instrument, which is described in the instrumentation section. We also calculated the percentiles for the same from the PSDs to give a measure of the variability in the aerosol properties.

**Response to Anonymous Referee #2**

Specific Comments

1. The ice number concentrations within the Sc clouds were generally a few per litre at most. This has now been stated in the text.

2. The low concentration is now stated as with Specific Comment #1

3. Double period corrected

4. We do focus on microphysical processes but have tried to provide a thorough overview of the different processes involved in the development of the Sc cloud layer. This has been detailed at the end of the introduction on page 4 lines 20 – 31.

5. We compared measurements from multiple different instruments in this paper and generally we found very good agreement between the measurements. We acknowledge the difficulty in instrument inter-comparisons but we found this to be a valuable thing to do as it helped us confirm that there is consistency in the measurements across a range of techniques.

6. Shattering on instrument inlets is a well-known phenomenon (e.g. Korolev et al. 2011). Some of the instruments were fitted with anti-shatter tips to reduce the number of shattered particles. However not all shattered particles can be removed in this way so we also used Inter-Arrival Time (IAT) Analysis to identify and remove particles with short inter-arrival times, which are likely produced by shattering on probe housing (e.g. Crosier et al. 2011).

7. As with Specific Comment #6 we carefully removed any shattering artefacts through IAT Analysis.

8. We have added a statement to the paper – 'We found agreement between the Total Water Content (TWC) Probes LWC and the Cloud Droplet Probe (CDP) LWC to be in good agreement for all cases. The $r^2$ values for these comparisons were 0.86, 0.86, 0.93 and 0.66 respectively. The lower $r^2$ value for the final case was due to an instrument issue that was identified using this inter-comparison approach. This further supports our thoughts in

Specific Comment #5 that inter-comparisons are useful for identifying potential problems. Below is an example figure of one inter-comparison between instruments.

[Figure]

**Figure. 1.** *An inter comparison time series between the CDP, TWC and LWC Sensor (left panel) and a scatter of CDP and LWC vs TWC LWC (right panel)*

9. Yes, the size threshold is pixels, the size value in the manuscript is a representation of the approximate particle size threshold using a pixel threshold of 50.

10. This has been altered in the manuscript.

11. The strength of a cold air outbreak is very sensitive to the time of year and also any given synoptic situation. Our inter-case variability was quite wide, but we saw similar outcomes in each case.

12. The profiles were selected by the flight pattern carried out by the aircraft. The approach was to fly saw tooth profiles through the Sc layers towards the open cellular region. In each case we tried to make as many measurements as possible within the cloud layers and the results of these profiles are presented for example in figures 6, 10 and 13. The variability is also presented to try and give some information on whether any trends were in excess of the variability.

13**.** If we were to compare cases as a function of altitude we don't feel normalised altitude would be an advantage as the altitude range the cloud layers spanned were similar in all cases.

14. The median value is quoted in the manuscript but we have added 25[th] and 75[th] percentile where appropriate for added information.

15. This information has been added

16. We aren't aware of a widely used index or strength indicator for the types of events described in the manuscript. The ECMWF ERA-5 Reanalysis products in figure 2 show some properties that can be used to infer the strength of a cold air outbreak. Mean Sea level Pressure (MSLP), 10m wind speed, 2 m temperatures and Sea Surface temperatures (SSTs) are all included in this figure.

17. The profiles were selected to try and represent the Sc cloud within the cloudy boundary layer and the changes that took place closer to the transition. Although we selected these by eye the data presented in figures 6, 10 and 13 show all data for all profiles to avoid any bias.

18. This has been changed in the text.

19. The figure has imagery superimposed on each particle size distribution from the individual cases, not just from a single case.

20. You are absolutely correct, we do not know for certain that dust is the INP active in this case, we can only make the suggestion based on our current understanding of studies into the different INP species active at different temperatures.

21. The imagery for each case is shown in figure 7 alongside the particle size distributions for the cases. However we have added a new figure 8 to provide more imagery from the cases.

22. We have included new figure 16 that shows data from the cases plotted on the same x axis (distance from breakup). We think this was a valuable suggestion and has helped show the cases that have some consistency with each other (cases 1 and 3) together with the case that had some differences. We have also improved the discussion section.

23. The Particles Size Distributions show the cases and the increasing amounts of larger drizzle sized particles closer to the breakup. We put the PSDs side by side so that the reader can compare the different cases.

24. The data and time captions have been added.

25. As in Specific Comment #23 we have the PSDs to represent the increasing drizzle size.